# Adaptive Syllable Training Improves Phoneme Identification in Older Listeners with and without Hearing Loss

**Annette Schumann** [1] and **Bernhard Ross** [1,2,*]

1   Rotman Research Institute, Baycrest Centre for Geriatric Care, Toronto, ON M6A 2E1, Canada
2   Department of Medical Biophysics, University of Toronto, Toronto, ON M5G 1L7, Canada
*   Correspondence: bross@research.baycrest.org; Tel.: +1-416-785-2500 (ext. 2690)

**Abstract:** Acoustic-phonetic speech training mitigates confusion between consonants and improves phoneme identification in noise. A novel training paradigm addressed two principles of perceptual learning. First, training benefits are often specific to the trained material; therefore, stimulus variability was reduced by training small sets of phonetically similar consonant–vowel–consonant syllables. Second, the training is most efficient at an optimal difficulty level; accordingly, the noise level was adapted to the participant's competency. Fifty-two adults aged between sixty and ninety years with normal hearing or moderate hearing loss participated in five training sessions within two weeks. Training sets of phonetically similar syllables contained voiced and voiceless stop and fricative consonants, as well as voiced nasals and liquids. Listeners identified consonants at the onset or the coda syllable position by matching the syllables with their orthographic equivalent within a closed set of three alternative symbols. The noise level was adjusted in a staircase procedure. Pre–post-training benefits were quantified as increased accuracy and a decrease in the required signal-to-noise ratio (SNR) and analyzed with regard to the stimulus sets and the participant's hearing abilities. The adaptive training was feasible for older adults with various degrees of hearing loss. Normal-hearing listeners performed with high accuracy at lower SNR after the training. Participants with hearing loss improved consonant accuracy but still required a high SNR. Phoneme identification improved for all stimulus sets. However, syllables within a set required noticeably different SNRs. Most significant gains occurred for voiced and voiceless stop and (af)fricative consonants. The training was beneficial for difficult consonants, but the easiest to identify consonants improved most prominently. The training enabled older listeners with different capabilities to train and improve at an individual 'edge of competence'.

**Keywords:** auditory rehabilitation; adaptive training; aging; hearing loss; speech-in-noise perception; phoneme identification

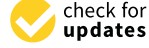



## 1. Introduction

Speech-in-noise (SIN) understanding becomes increasingly challenging at an advanced age and severely impairs verbal communication [1]. An impaired communication ability is, at first, an individual problem; however, it impacts mental, cognitive, and general health and becomes a significant burden for the healthcare system and society as a whole, given the large prevalence of auditory impairment in the aging population. Substantial progress has been made in mediating hearing loss through hearing aids, cochlear implants, and other assistive technology. However, listening and understanding require brain function to interpret speech sound and extract its meaning. When audibility is restored after a period of deprivation from auditory input, the brain spontaneously learns how to use the acoustic information. However, listening training could facilitate the efficient use of restored hearing for speech understanding. Moreover, listening training could help to develop strategies to extract speech information when acoustic cues are not fully available, distorted, or masked by concurrent noise. Therefore, listening training in auditory rehabilitation aims

to restore SIN recognition skills and enhance communication abilities in older adults [2,3]. However, characterizing the benefits of auditory training [4] and finding the optimal training paradigm [5] are still matters of current research. In our study, we addressed questions about an optimal training material and the training procedure could be designed to accommodate older adults within a wide range of hearing and listening abilities.

Training-related improvement in speech recognition utilizes the principles of perceptual learning [6,7]. First, perceptual learning is specific to the trained material, and transfer has been predominantly observed within the same stimulus category [8]. Therefore, we grouped trained syllables according to their phonetic similarities within a specific stimulus category to improve discrimination abilities for phonetic details. Second, learning seems optimal when listeners perform close to the individual limit of difficulty or competence [9–11]. Therefore, we designed the training as an adaptive procedure by adjusting each participant's signal-to-noise ratio (SNR).

Listeners in a noisy environment may confuse a speech token for a similar sound. For example, mistaking 'someday' for 'Sunday' could cause a grievous communication error. Such difficulties in speech intelligibility depend on the relationship between the intensities of the speech sound and the concurrent noise, i.e., the SNR, rather than the absolute noise level [12]. In some cases, a single dB increase in SNR provided up to 20% better speech recognition in older normal-hearing listeners [13].

Hearing loss complicates SIN understanding. Presbycusis, the aging-related changes in the peripheral and central auditory systems [1,14–16] elevates hearing thresholds and impacts speech discrimination abilities. For example, impaired outer hair cell function reduces cochlear amplification, causes high-frequency hearing loss, and widens frequency selective channels—all contributing to SIN deficits. Moreover, the loss of auditory nerve synapses impairs sound perception [17] and cognitive decline affects speech processing [18–20].

Presbycusis affects SIN understanding in a complex manner. Age-related hearing loss is most substantially expressed at higher frequencies [21] and impacts high-frequency speech cues crucial for consonant discrimination [22]. In contrast, vowel perception seems well-preserved [23,24]. A gradual increase in high-frequency hearing loss over time leads to compensatory neuroplastic changes in the auditory cortex [25]. Then, the listener can utilize low-frequency temporal cues for phoneme identification, such as the vowel duration [26]. However, if multi-talker noise masks this low-frequency information, the listener relies on higher-frequency speech cues. If the noise masks a perceptually relevant acoustic-phonetic feature of a speech sound, it can be confused with another similar sound. Such confusion results in a significant variability of consonant recognition scores [27]. Therefore, we expected the largest confusion for the stimuli matched for phonetic similarity with a substantial range of inter-individual variability.

Generally, listeners with hearing loss require a larger SNR to achieve the same speech intelligibility as their normal-hearing peers [28,29]. Therefore, the SNR was the primary outcome measure for the training benefits in our study, indicating that participants could understand speech at a lower SNR or improve word recognition within the same SNR level. Moreover, the SNR is most relevant because hearing thresholds alone cannot predict older listeners' SIN deficits, which depend on the SNR [30–32].

Older listeners suffer from impaired consonant identification, even when using hearing aids [33–35]. Although the speech signal is fully audible, intelligibility might not be restored entirely [31,36], and deficits remain at a relatively high SNR [18]. The common complaint, 'I can hear you, but I cannot understand what you said,' indicates SIN deficits in audible speech. While restoring audibility is the foremost important goal of auditory rehabilitation, further listening training may help alleviate the remaining difficulties in SIN understanding.

A current opinion is that listening training improves how speech is encoded and processed in the central auditory system, and that repetitive, active exercises enable individuals to make better use of existing patterns [37–39]. Training-induced performance improvements are driven by neuronal reorganization within the auditory modality [9,11,40] and restore the utilization of high-frequency consonant information in older listeners with

hearing loss [41]. In phoneme-based training, perceptual learning remediates consonant-identification deficits by reducing phonetic confusion and enhancing the listener's ability to identify previously difficult syllables out of noise [42]. Perceptual learning impacts multiple levels, such as facilitating sensory representation and improving higher-level processing [43,44]. The higher-level component involves how the listener interprets the sensory signal for decision making that determines the behavioural response [45]. If embedded in an active task, perceptual learning can occur after an intervention as short as one hour [46,47].

Phoneme-based listening training aims to increase the ability to distinguish subtle acoustic-phonetic speech cues in syllables or monosyllabic words [48]. The training focuses on a listener's phonetic-phonological speech processing skills by excluding otherwise helpful semantic context information. Therefore, phoneme-based training often employs nonsense syllables, constructed as consonant–vowel (CV), vowel–consonant (VC), or consonant–vowel–consonant (CVC) syllables. Examples of such training material are the CUNY Nonsense Syllable Test [22] and the California Syllable Test (CaST) [49]. Initially, these materials were developed for measuring consonant and vowel identification performance in noise in older listeners with hearing loss and were later used for training.

Reports about the outcomes of phoneme-based training are not entirely consistent. Several studies demonstrated beneficial effects. For example, after 40 hours of at-home training on nonsense syllables presented in speech-spectrum noise [41], 94% of older hearing aid users (mean age 70 years, using hearing aids for longer than eight months) could identify consonants at lower SNR than before the training. In their study, several consonants required a high SNR, and the training benefits were the largest for these difficult-to-identify stimuli. In another study with unaided and aided listeners, identification performances for syllables presented with constant SNR increased after an eight-week phoneme-based training [38]. Stimuli in their study were available in two male and two female voices, one male and one female voice were randomly assigned for the training in each individual, and all voices were tested after the training. The training benefits for the untrained voices, although smaller than for the trained voices, indicated generalization.

However, phoneme-based training was less efficient in other studies [5,48,50,51], and the optimal paradigm for this training approach still needs to be identified.

The selection of the stimulus material is crucial for the training outcome. Previous findings informed our choice of syllables that perceptual learning is specific to trained material, and the transfer of learning occurs primarily within the same stimulus category [8]. Task complexity impacts generalization. For example, in training, using a variety of speech stimuli, generalization was observed more commonly to identify contextually rich meaningful sentences or highly discriminable words, whereas learning difficult speech cues was more specific [52].

Moreover, limiting the stimulus variability seems essential. For example, perceptual learning in the visual domain depends on the constancy of properties of the learned object, whereas sensory representations could vary [53].

The training setup for this study followed three design principles. First, we created sets of highly confusable stimuli based on evidence that phoneme-based training can reduce confusion among specific consonant pairs [41]. Therefore, we grouped the syllables into sets containing voiced or voiceless consonants of similar phonetic features. The training task resembled the minimal pair identification task used in previous studies [54]. Second, the complex CVC syllables allowed to examine the impact of the consonant position within the syllables on the training outcome. The participants identified a consonant at the onset or coda position within the CVC syllables. Previous studies found that consonant identification was more difficult and improvement was the largest for the coda condition [41,55]. Accordingly, we predicted more extensive benefits for identifying the more difficult coda consonants. Third, participants matched the stimulus with three alternatives selected from within the stimulus set. Using closed-set paradigms was successful in other phoneme-

based training studies, such as within the n-alternative forced-choice (AFC) training of phonological contrasts [10,56].

We implemented the training and testing paradigms as staircase procedures for adjusting the noise level according to individual performance. Individualizing the staircase paradigm to train the participants' abilities could encourage their continued engagement by being challenged but not frustrated. Alternatively, if the training would operate at a fixed SNR, high-performing individuals may not improve further because they are already trained at the ceiling level.

An adaptive staircase procedure adjusts the SNR according to a fixed supra-threshold performance level [34,57] which has already been used in phoneme-based training for older listeners [41,56]. The number of subsequent correct or incorrect responses leading to a change in difficulty determines the performance level. For example, participants in three-up and one-down procedures performed at 79% accuracy [58].

The staircase concept was borrowed from psychometrics, but the similarities end there. While the SNR converges to a threshold within small step sizes in a psychometric test, we expected considerably more variability within the staircase training procedure. Generally, a staircase procedure requires a monotonous relationship between stimulus and performance levels to converge to a threshold [57]. Stimulus variability could compromise the asymptotic tracking behaviour [59]. However, a wide SNR range was required for equal identification performance when using complex CVC syllables [55]. Preparing the stimuli for their training study, Woods et al. [41] divided the consonants into groups based on their identifiability after minimizing the inter-stimulus differences by adjusting each consonant's SNR. Such an approach was not appropriate for our study focusing on the ability to distinguish syllables with similar phonetic features and varying noise. Therefore, we decided to present the individual stimuli at same intensity and accepted heterogeneity within the stimulus sets. Thus, the 'loudness' of noise and target speech fluctuated and challenged the clear perception of essential speech cues. This approach closely reflected the everyday listening situation. Effectively, the training procedure adjusted the SNR in the mean according to the trainee's abilities.

We expected participants to improve their accuracy and require a lower SNR for syllable identification. However, the type and magnitude of training benefits may depend on individual hearing abilities. Moreover, we expected performance differences between the syllable sets and onset and coda identification. We analyzed the individual performances throughout the training and with pre–post-training tests. Additionally, we compared the performances and training effects for the individual consonants within a syllable set, which informs how well phonological similarity equated to difficulty in the task. Lastly, we analyzed the training performances in subgroups of different hearing loss and SIN deficits, which informed how well the adaptive training paradigm is suitable for a heterogeneous population.

## 2. Materials and Methods

### 2.1. Study Design and Procedure

The syllable training was part of a crossover-design study with two training programs. One was a phoneme identification task to improve syllable recognition in noise; the other trained sentence recognition in noise, using keyword identification in repeated sentences. Here, we report the syllable training only. Syllable recognition was tested before and after training and observed during each session. Moreover, speech-in-noise tests were applied to determine whether syllable training could be generalized to sentence recognition.

### 2.2. Participants

Fifty-eight adults between 60 and 90 years of age were recruited from the Rotman Research Institute volunteer database and in-person during information sessions with the communities. The participants reported being healthy with no history of neurological or psychiatric disorders. Participants gave written consent after they were informed about the

nature of the study and received an honorarium for participation. The study protocols conformed to the World Medical Association Declaration of Helsinki [60] and were approved by the Research Ethics Board at Baycrest Centre for Geriatric Care (REB 18-35).

The compliance rate for the training was high. Two participants retracted because of health problems, and four could not complete all testing sessions due to time commitment. Therefore, data from 52 participants (28 female) with an average age of 71 years (SD: 7) were analyzed. Twenty-four participants used hearing aids for 1–20 years (M: 6.6, SD: 5). Most used their hearing aids regularly, while two reported wearing their devices occasionally.

Before the training, all participants were screened with the Montreal Cognitive Assessment (MoCA) [61] to determine possible mild cognitive impairment. Forty-three out of 52 scored in the normal range (>=26), eight participants scored slightly lower between 25 and 23, and one participant scored 18. However, no participant was excluded, and the MoCA score was used as a between-participant factor for data analysis. Participants with borderline mild cognitive impairments were part of the target population for SIN training. Again, one goal was to see whether adaptive training could accommodate such a heterogeneous participant population.

### 2.3. Hearing Assessment

Before the first training sessions, hearing thresholds for both ears were tested with pure tone audiometry using the modified Hughson–Westlake procedure at the octave frequencies between 250 Hz and 8000 Hz. We used the four-frequency pure tone average (PTA) definition, including 500 Hz, 1000 Hz, 2000 Hz, and 4000 Hz, as commonly used for studying speech perception [18].

SIN understanding was tested with the Quick Speech-in-Noise Test (QuickSIN) [62] and the Bamford–Kowal–Bench Speech-in-Noise Test (BKB-SIN) [63]. In QuickSIN, six sentences with five keywords were presented binaurally at 70 dB normal hearing level with concurrent four-talker babble noise at SNR between 25 dB and 0 dB, decreasing in 5 dB steps. The test sentences from the Harvard corpus [64] were phonetically balanced and provide few semantic context cues. In the BKB-SIN test, sentences with three keywords were presented binaurally at 70 dB normal hearing level, embedded in a four-talker babble noise decreasing from SNRs 21 dB to −6 dB in 3-dB steps [65]. The BKB-SIN sentences contain a rich semantic context and high redundancy and are often used for older listeners when the QuickSIN is too challenging. QuickSIN and the BKB-SIN scored the number of correctly identified keywords at the different SNR levels and calculated the SNR for 50% correct word identification. The difference compared to the SNR50 in the normal population is termed SNR loss. Instead, we used the term SIN loss [66].

All hearing tests were performed in a soundproof booth using a clinical audiometer (GSI61, Grason Stadler, Eden Prairie, MN, USA) and ER-3A sound transducers (Etymotic Research, Elk Grove Village, IL, USA) connected to the participants' ears with 20 cm plastic tubing and foam earpieces.

### 2.4. Stimuli

We selected syllable stimuli from the California Syllable Test (CaST) [67]. The corpus is a rich set of 9600 mostly nonsense syllables [49,55]. We chose these stimuli to train the onset and coda identification of consonants in syllables. Twenty-one consonants were embedded in three different vowel combinations (/ɑ/, /i/, /u/). Natural speech variability was considered using multiple versions of each stimulus uttered by two male and two female adult American English speakers. Examples of the waveforms and spectra of male utterances of 'maat' and 'maash' are illustrated in Figure 1a,b. The spectrograms show two phonemes' spectral patterns with the typical stimulus onset, the vowel, and stimulus offset differences. The period of silence before the consonant /t/ in 'maat' represents the closed vocal tract before the moment of the 'explosive' release within a short time. In contrast, the fricative /sh/ in 'mash' shows no silent period but a long-lasting burst of high-frequency sounds. The duration of the syllables was between 600 ms and 900 ms. The noise was

stationary Gaussian noise, filtered according to the spectral profile of speech (Figure 1c). The noise began 200 ms before stimulus onset and lasted for 1100 ms, including 50 ms cosine slopes at onset and offset.

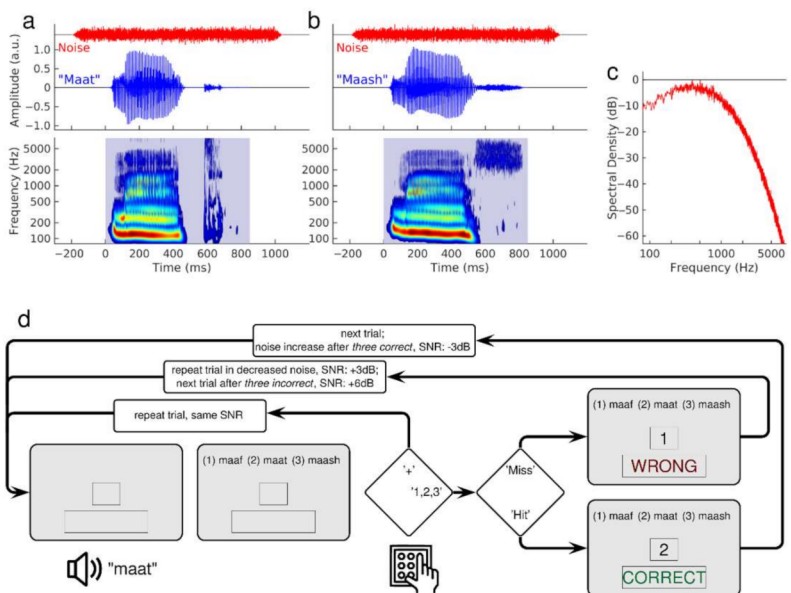

**Figure 1.** Stimuli and training paradigm. (**a**): Time series of the syllable 'Maat' and the concurrent noise show the temporal relationships between stimuli and noise. The spectrogram of the syllable sound illustrates the distinct spectral patterns during the syllable onset, the vowel, and the coda. (**b**): Time series and spectrogram for the syllable 'Maash' show the different spectral pattern of the coda compared to 'Maat' (**c**): Power spectrum of the speech-like filtered Gaussian noise. (**d**): Flow chart of one trial of the training procedure. The example shown is from a coda syllable set. Participants had the options to request a stimulus repetition or selected the written syllable best matching with the sound stimulus. Visual feedback was provided and the SNR for the next trial was adjusted according to the participant's response.

The CVC syllables were sorted according to their phonetic similarity into sets of stimuli presented within a training block. Previous studies using confusion matrices [22,34,68–72] showed that some phonemes were more often misidentified than others. Confusion between voiced and voiceless phonemes such as /b/ and /t/ was low. However, consonant misidentification frequently occurred within the manner or the place of consonant articulation, such as confusion between higher-pitched stops /p/ and /t/ or in nasals between the labial /m/ and the alveolar /l/.

We created three syllable sets according to voiced or voiceless consonants. The sets contained either stop, fricative and affricative consonants, the latter two labelled as (af)fricative, or nasal, liquid, and vibrant consonants. We combined stops and (af)fricatives into one set to find a compromise between the consistency of stimuli features, practicable set size, and the number of sets. Each consonant set was presented in the onset task for the identification of the leading consonant only and the coda task for the identification of the ending consonant. Thus, the training involved a total of six consonant conditions.

The set C1 contained seven voiced stop and (af)fricative consonants (/b/, /d/, /ð/, /g/, /ʤ/, /v/, /z/), and the set C2 contained eight voiceless stop and (af)fricative consonants (/p/, /t/, /θ/, /k/, /ʧ/, /ʃ/, /f/, /s/). C3 contained four voiced mainly nasal and glides consonants plus one glottal consonant (/m/, /n/, /l/, /r/ + /h/) in the onset condition and the same four voiced mainly nasal and glides consonants plus one voiced nasal consonant (/m/, /n/, /l/, /r/, + /ŋ/) in the coda condition. We combined multiple versions of the consonants with three different vowels (/ɑ/, /i/, /u/) to guarantee high stimulus variation within a set.

We presented the syllables in orthographical writing instead of the phonetic inventory for easier readability. Hence, we replaced /ð/ with /DH/, /ʤ/ with /j/, /ɵ/ with /th/, /ʧ/ with /ch/, /ʃ/ with /sh/, and /ŋ/ with /ng/. The phonemes are also presented orthographically in the following parts of the paper.

*2.5. Adaptive Training Procedure*

Over the two-week training period, participants performed five training sessions. Each session consisted of six blocks. In each block, participants trained consonants from one of the six sets for seven minutes. On average, 90 trials were trained in a block, depending on the participant's pace. Each block contained a minimum of 45 trials, and the maximum was 145. Forty-three participants trained at least 70 trials per block and 14 more than 110 trials per block. Most participants could complete the training within 45 min. Some participants proceeded slower and required up to 60 min per session. Phoneme identification performance was tested on a day before the first training session and another day after completing the training program.

Participants in a classical n-AFC experiment listen to a sequence of n stimuli and identify the one containing the target feature. However, in our study, participants listened to one syllable and matched it to three orthographically presented alternatives. The training used a three-up, one-down staircase implemented in the Matlab Psychoacoustics toolbox [73] to adapt the SNR to individual performance in a modified 3AFC paradigm.

Immediate feedback and stimulus repetition reinforced learning in previous studies [40,74]. Therefore, visual feedback was provided after each response. Additionally, we offered the option of multiple repetitions of the syllable sound before the participant responded. Moreover, in the case of an incorrect identification, the same sound stimulus was presented again together with the same set of orthographic alternatives. The trainee was encouraged to use stimulus repetitions and multiple attempts to aim at a correct response. Thus, the motivational level should stay high during the challenging training task [75,76].

Figure 1d shows a flowchart of the training for the coda identification task with the voiceless C2 consonant set. Each trial started with an auditory presentation of the syllable in noise, followed by an orthographic representation of three alternative syllables on the computer screen. The participant chose the syllable best matching the acoustic presentation by responding via keypad numbers 1, 2, or 3. To request a stimulus repetition at the same SNR, the participant pressed the '+' key. After three trials with consecutive correct identifications, the next stimulus was presented at a 3 dB lower SNR. With each incorrect response, the syllable was repeated with less background noise by increasing the SNR by 3 dB. After three wrong answers, a new stimulus was presented at the most recently adjusted SNR. The step size changed to 2 dB after 20 reversals. The easiest level of SNR was 20 dB, which was assumed to be above the threshold for most of the consonants based on previous results on SNR estimation for the CaST syllables [49,55]. The most difficult SNR level was −10 dB.

*2.6. Phoneme Identification Tests*

Phoneme identification was tested before and after the training. The stimuli were taken from the training material. The test used the same staircase procedure as the training itself (Figure 1d); however, without feedback and the option of stimulus repetition. After each response, a new syllable was presented with the SNR adjusted by a step size of 2 dB, increasing after incorrect and decreasing after correct answers. In each test session, all six syllable sets were tested. A test was completed after eight reversals, which resulted in 80–120 trials for each syllable set. While the training paradigm allowed the trainee to perceive the information necessary for correct stimulus identification, the test paradigm challenged speech perception difficulties to measure the performance and possible training effects. Therefore, the easiest SNR level was set at 10 dB, close to the estimated 12 dB necessary for consonant identification in older listeners with hearing loss for the CaST [48]. The most difficult SNR level was −10 dB. A testing session lasted 60–70 min.

The syllable testing and training were performed in a quiet testing room in the Rotman Research Institute with ongoing assistance for each session. The stimuli and noise were presented through a speaker (Energy Excel Audiophile Loudspeaker System) at a one-meter distance in front of the participant. The stimulus intensity was controlled with an audiometer and calibrated with a sound-pressure level meter (824, Larson-Davis, Depew, NY, USA). Hearing aid listeners performed training and testing with their own devices set to their everyday listening program. The speech sounds were presented at 65 dB SPL for the participants within the normal hearing range (PTA < 25 dB). The presentation level could be individually adjusted to comfortable loudness up to 75 dB for participants with moderate hearing loss and no hearing devices. The stimulus intensity was 70 dB for most of the participants. The individual loudness level was kept constant for all training sessions.

### 2.7. Training Benefits and Test Outcome Measures

Phoneme identification tests and training were performed separately in blocks for each syllable condition. Accuracy and SNR were measured as the mean across all trials in an experimental block. First, for each block, a confusion matrix was created with a size $7 \times 7$ for the consonant set C1, $8 \times 8$ for C2, and $5 \times 5$ for C3 using the empirical probability h(r|s) that a consonant /r/ was identified given the test stimulus /s/ had been presented. The confusion matrix allowed for an analysis of the types of misidentifications and whether those changed during the training. We calculated the d-prime sensitivity measure [77], only considering the hit rate and false-alarm rates regardless of the type of confusion for every consonant in each of the six sets.

The hit rate H was defined as the empirical probability that the participant responded with the consonant /c/ when the stimulus /c/ was given, H = p(c|c). The false-alarm rate F was the probability of the response /c/ when a stimulus other than /c/ had been presented, F = p(c≠c). Then, d-prime was calculated as $d' = z(H) - z(F)$, in which z is the inverse of the normal probability distribution function. We calculated the mean d-prime and SNR for the three consonant sets for onset and coda positions.

### 2.8. Statistical Analysis

Phoneme identification was analyzed with repeated-measures ANOVA for d-prime and SNR with the 'factors', 'consonant set' (C1, C2, C3), 'syllable type' (onset and coda) and 'session'. The factor' session' had five levels for the training data and two levels for the pre- and post-training tests. The effect of peripheral loss and cognitive decline on the training outcome was considered by including PTA and MoCA scores as a between-participant factor. All ANOVA analyses were performed in R [78].

We analyzed how the level of individual hearing loss impacted the training benefit. For this analysis, we explained pre- and post-training SNR and d-prime by linear models of PTA for each of the six stimulus conditions. Moreover, we created three groups (low-PTA, mid-PTA, and high-PTA) by stratifying participants into subgroups using the 25% and 75% percentiles of their PTA and compared the training benefits between the PTA groups.

Lastly, we analyzed how training outcomes depended on the different consonants within each set. For this analysis, six separate repeated-measure ANOVAs for d-prime and SNR were performed with the 'factors', 'consonants' and 'session'. The factor 'consonants' had either five, seven, or eight levels.

### 3. Results

### 3.1. Hearing Loss and SIN Loss in Older Age

The two-way ANOVA for the audiometric thresholds with the factors 'ear' and 'frequency' (six levels) revealed a main effect of 'frequency' (F(5,586) = 28.7, $p < 0.0001$). As expected, thresholds in older adults were elevated at higher frequencies. The ANOVA did not show an effect of 'ear' (F(1,586) = 0.33) nor an 'ear' × 'frequency' interaction (F(5,586) = 0.17). Therefore, the PTA hearing thresholds were averaged across ears.

In Figure 2a, the audiograms were averaged across decades of participant's age between their 60s and 90s years to illustrate the characteristic high-frequency loss. Individual PTA values, ranging from 8.1 dB to 66.9 dB (m: 34.1 dB, sd: 15.6 dB), are shown in relationship to the participant's age in Figure 2c. PTA was highly correlated with age ($R^2$ = 0.33, F(1,51) = 24.4, *p* < 0.0001).

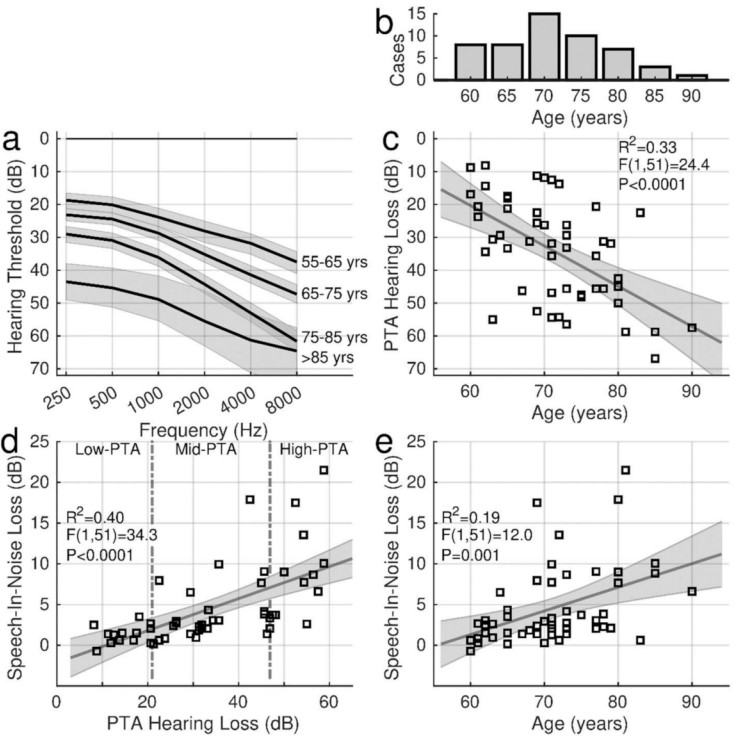

**Figure 2.** Hearing thresholds and speech-in-noise (SIN) loss. (**a**): Mean audiograms for participants between their 60s and 90s, averaged across right and left ears. The shaded areas indicate the standard error of the mean. (**b**): Distribution of participants across the ages. (**c**): Individual pure-tone-average (PTA) hearing loss versus age, modelled by linear regression. The grey shaded area indicates the confidence interval for the linear model. (**d**): Individual SIN loss obtained with QuickSIN versus age. (**e**): SIN loss versus PTA. The dashed lines mark the ranges for three PTA subgroups.

The SIN performance was determined before the training as the SIN loss in QuickSIN and BKB-SIN. In the group mean, the pre-training QuickSIN loss of 4.83 dB (95% CI [3.44, 6.21]) was larger than the BKB-SIN loss (3.12 dB, 95% CI [2.53, 3.70]). SIN loss was correlated with hearing loss (Figure 2d), and this relationship was steeper for QuickSIN (0.20 dB per dB hearing loss, $R^2$ = 0.40, F(1,50) = 34.4, *p* < 0.0001) than for BKB-SIN (0.07 dB per dB, $R^2$ = 0.30, F(1,50) = 21.7, *p* < 0.0001). Again, the SIN loss was correlated with age (Figure 2e). The SIN loss increase with advancing age was more pronounced for QuickSIN (0.30 dB per year $R^2$ = 0.19, F(1,50) = 12.0, *p* = 0.001) than for BKB-SIN (0.13 dB per year, $R^2$ = 0.20, F(1,50) = 13.1, *p* = 0.0007).

QuickSIN was also moderately correlated with the MoCA scores ($R^2$ = 0.09, F(1,50) = 4.77, *p* = 0.034) while PTA was not correlated with MoCA (F(1,50) = 1.1, *p* = 0.3, n.s.).

Despite the correlations between age, hearing loss, and SIN loss, individual hearing abilities varied considerably, and variability was most prominent in older individuals with more extensive hearing loss. While SIN loss and more severe hearing loss were more pronounced in older age, several older individuals with hearing loss showed only a few dB of SIN loss. For example, the SIN loss was less than 5 dB for individuals with a PTA below 20 dB (Figure 2d) or below age 65 (Figure 2e).

The quartiles of PTA were used as a categorical variable for describing how the training outcomes depended on hearing abilities. The vertical dashed lines in Figure 2d separate three PTA groups. The PTA range for the 13 individuals in the low-PTA group was 8 dB–21 dB (m = 15.0 dB). The PTA in the mid-PTA group (n = 26) ranged between 21 dB and 47 dB (m = 33.6 dB) and between 47 dB and 67 dB (m = 54.4 dB) in the high-PTA group. Thirteen of the mid-PTA participants and eleven participants in the high-PTA group used hearing aids.

### 3.2. Training Outcome Measures

Figure 3 illustrates how the mean SNR and accuracy were obtained as the outcome measures. Sometimes, the SNR required for identification varied between the subsequent stimuli more than the step size of the staircase. Therefore, the course of the SNR level during a block of approximately 100 trials revealed more volatility than one would observe in a typical psychometric test (red graph in Figure 3a). Moreover, within the given SNR range, some participants did not reach sufficient accuracy, for a converging procedure, resulting in an SNR close to the ceiling (blue graph in Figure 3a). As an outcome measure, the SNR was calculated as the mean across all trials, visualized with horizontal lines in Figure 3a. The figure also provides an example of a participant, requiring different SNR levels for different consonant sets, C1 and C2.

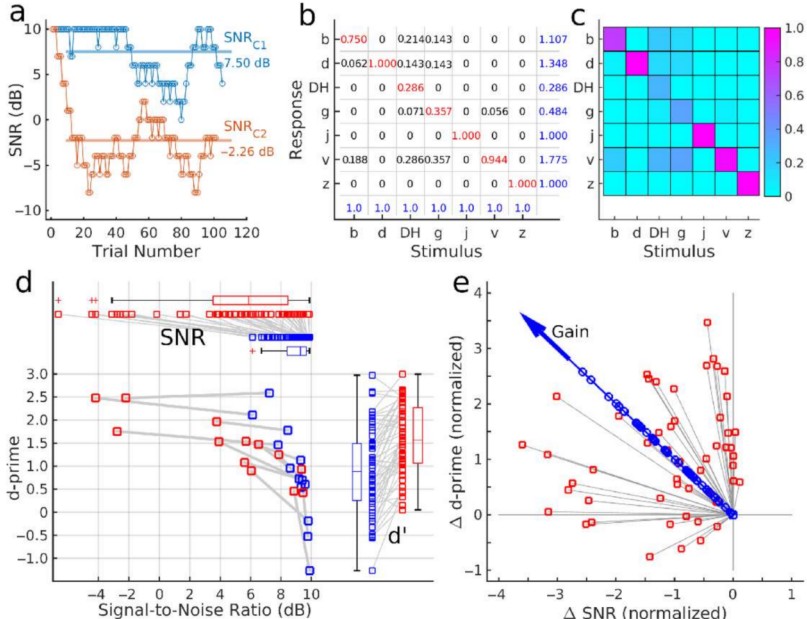

**Figure 3.** Outcome measures of the syllable training. (**a**): Example of the course of SNR level during test blocks with the consonant sets C1 and C2. The horizontal lines indicate the mean SNR across all trials. (**b**): An accuracy measure was obtained from confusion matrices. The example shows the empirical response probabilities of an individual participant. Blue numbers in the bottom row are the sum within the column (stimulus probability = 1.0). Blue numbers in the right column are the sum of response probabilities across the column. (**c**): Color map of the confusion matrix. (**d**): Visualization of SNR and d-prime before and after the training of a single point in the d-prime versus SNR data plane. Participants with low d-prime pre-training increased their accuracy after the training while the SNR remained high. Participants with high pre-training d-prime maintained the high accuracy level while decreasing the SNR. The box plots on the right indicate d-prime changes and the upper box plots SNR changes after the training. (**e**): A gain measure was obtained as the first principal component of the normalized d-prime and SNR differences between the pre- and post-training tests.

The panels b and c in Figure 3 illustrate the numerical values and a color map of an example confusion matrix which was used to calculate the phoneme identification accuracy.

In each trial, participants correctly identified or misidentified the presented phoneme. The empirical response probabilities h(r|s) of responding with /r/ given the stimulus /s/ were the elements of the confusion matrix (Figure 3b). The blue numbers in the bottom row are the sum of elements in each column, equal to 1.0, the total probability of presenting the stimulus /s/. Blue numbers on the right represent the sum of row elements, indicating the frequency of responding with a particular consonant. In this example, the participant responded more frequently with the consonant /v/ at 1.775 than /DH/ at 0.286.

The confusion matrix's main diagonal contained the probability of correct identification, i.e., the hit rate. In the example, the participant correctly identified the consonants /d/, /j/, and /z/ according to the value 1.0 in the diagonal. The sum of all other matrix elements in a row equals the probability of falsely identifying a particular consonant, i.e., the false-alarm rate. Hit rates and false-alarm rates were used to calculate the d-prime values of the measures of accuracy.

The two outcome measures were visualized as a single data point in the SNR-d-prime data plane. The pre- and post-training test results in Figure 3d illustrate that performance changes between pre- and post-training tests differed between individuals. Participants who performed with low accuracy at the 10-dB SNR limit of the test (left lower corner of Figure 3d) improved their accuracy but did not advance to phoneme identification in lower SNR after the training. Other participants who performed with higher d-prime values before the training could correctly identify phonemes at a higher noise level (i.e., a lower SNR) after the training. The individual d-prime values, shown on the right side of Figure 3d, demonstrate a substantial range of inter-individual performance differences. However, the boxplots indicate an increase in the median d-prime value between the two test points. Individual SNR values and boxplots at the top panel of Figure 3d demonstrate that most participants performed the test at lower SNR in the post-training test than before the training. The performance during the five training sessions was analyzed in the same way.

The training benefits in SNR and d-prime were combined into a single gain measure (Figure 3e). The pre–post differences Δd-prime and ΔSNR were normalized by their standard deviations and submitted to principal component analysis. The signed magnitude of the first principal component was used as a normalized measure of the training benefit.

### 3.3. Effect of Training

The ANOVA for d-prime observed during the five training sessions with the factors 'consonant set' (C1, C2, and C3), 'syllable type' (onset, coda), and 'session' revealed a main effect of 'session' ($F_{(4,204)} = 5.38$, $p = 0.0004$); d-prime increased throughout the training. The effect of 'session' was also significant in the same design ANOVA for the SNR ($F_{(4,204)} = 23.4$, $p < 0.0001$); participants performed at lower SNR at the end of the training than at the beginning. The absence of 'session' interactions indicated that the SNR homogenously decreased for all stimuli.

Pre- and post-training tests confirmed the training-related pattern of increased phoneme identification accuracy and a decrease in the required SNR. The ANOVAs for d-prime and SNR (Table 1) with the within-participant factors 'consonant set', 'syllable type', and 'session' (pre–post) revealed the main effect of 'session' (d-prime: $F_{(1,50)} = 77.1$, $p < 0.0001$, SNR: $F_{(1,50)} = 73.6$, $p < 0.0001$).

The grand mean d-prime and SNR values (Figure 4) showed noticeably different levels between the training and test sessions. The group mean d-prime values during the training were relatively high because several features, such as repeating the stimuli, made the syllables highly recognizable. Thus, accuracy was almost constant across the training sessions. However, the participants improved their ability to listen at a lower SNR. Although the SNR was relatively high during the training. During the more challenging tests, participants performed at a lower SNR and consequently lower d-prime. Nonetheless, both measures showed substantial training benefits in the group mean.

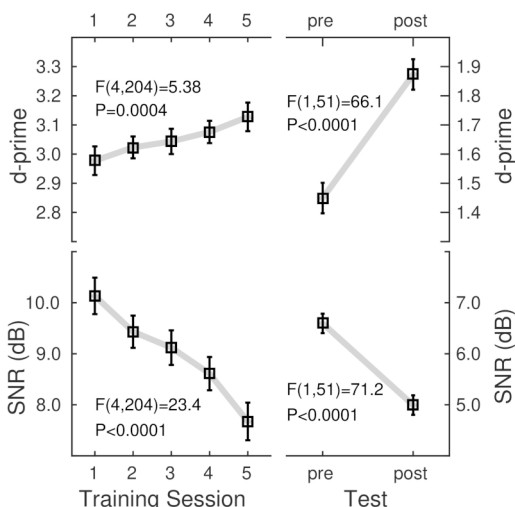

**Figure 4.** Grand mean d-prime and SNR changes across the five training sessions and between pre- and post-training tests. F-statistic and *p*-values are the main effects of the session determined by ANOVA with repeated measures. Error bars indicate the 95% confidence intervals. Note the different y axis ranges for the training and tests.

**Table 1.** Repeated-measure ANOVA for d-prime and SNR with the between-participant factor 'PTA' and within-participant factors 'session' (pre- and post-training), 'consonant set' (C1, C2, C3), and 'syllable type' (onset and coda) and ANOVA for the normalized gain. Significant effects ($p < 0.05$) are in bold.

| Effect | D-Prime | | SNR | | Gain | |
|---|---|---|---|---|---|---|
| PTA | $F_{(1,50)} = 48.3$ | ***p* < 0.0001** | $F_{(2,50)} = 68.1$ | ***p* < 0.0001** | $F_{(1,50)} = 0.90$ | *p* = 0.35 |
| Session | $F_{(1,50)} = 77.1$ | ***p* < 0.0001** | $F_{(1,50)} = 73.6$ | ***p* < 0.0001** | | |
| Consonant Set | $F_{(2,100)} = 52.9$ | ***p* < 0.0001** | $F_{(2,100)} = 46.9$ | ***p* < 0.0001** | $F_{(2,100)} = 19.1$ | ***p* < 0.0001** |
| Syllable Type | $F_{(1,50)} = 57.6$ | ***p* < 0.0001** | $F_{(1,50)} = 86.3$ | ***p* < 0.0001** | $F_{(1,50)} = 1.12$ | *p* = 0.29 |
| Session × Consonant Set | $F_{(2,100)} = 18.8$ | ***p* < 0.0001** | $F_{(2,100)} = 8.09$ | ***p* = 0.0006** | | |
| Session × Syllable Type | $F_{(1,50)} = 1.42$ | *p* = 0.24 | $F_{(1,50)} = 8.10$ | ***p* = 0.006** | | |
| Consonant Set × Syllable Type | $F_{(2,100)} = 103$ | ***p* < 0.0001** | $F_{(2,100)} = 79.2$ | ***p* < 0.0001** | $F_{(2,100)} = 2.12$ | *p* = 0.13 |
| PTA × Session | $F_{(1,50)} = 9.41$ | ***p* = 0.0035** | $F_{(1,50)} = 2.75$ | *p* = 0.10 | | |
| PTA × Consonant Set | $F_{(2,100)} = 2.04$ | *p* = 0.13 | $F_{(2,100)} = 10.9$ | ***p* < 0.0001** | $F_{(2,100)} = 0.11$ | *p* = 0.75 |
| PTA × Syllable Type | $F_{(1,50)} = 9.95$ | ***p* = 0.003** | $F_{(1,50)} = 20.4$ | ***p* < 0.0001** | $F_{(2,100)} = 1.7$ | *p* = 0.18 |

*3.4. Effects of the Stimulus Sets*

The ANOVA for the pre- and post-training test d-prime (Table 1) showed the effects of a 'consonant set' ($F_{(2,100)} = 52.9$, $p < 0.0001$) and 'syllable type' ($F_{(1,50)} = 57.6$, $p < 0.0001$) indicating that the accuracy was different for the various stimulus conditions (Figure 5a). The grand mean d-prime for C2 was higher than for C1 ($t_{(102)} = 6.03$, $p < 0.0001$) and for C3 ($t_{(102)} = 11.21$, $p < 0.0001$). Similarly, the ANOVA for SNR showed the effects of the 'consonant set' ($F_{(2,100)} = 46.9$, $p < 0.0001$) and 'syllable type' ($F_{(1,51)} = 86.3$, $p < 0.0001$), indicating that the participants required noticeably different SNR values for the different stimuli. The SNR values ranged between a low 4.2 dB for the onset condition of the C3 set and a high 8.3 dB for the coda consonants in the C1 set (Figure 5b).

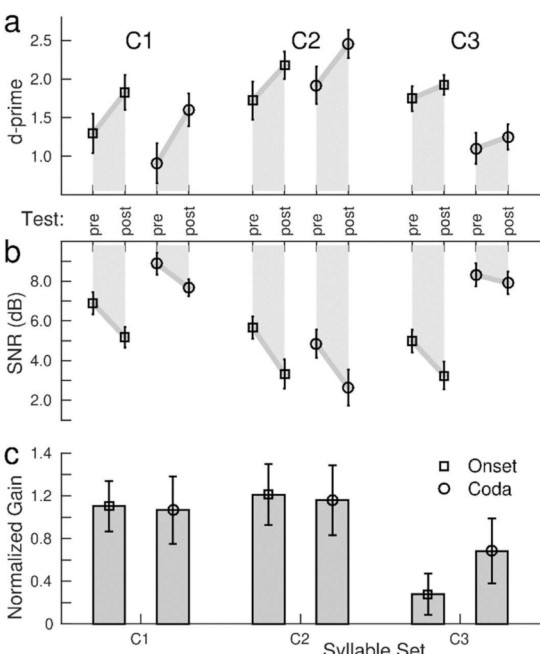

**Figure 5.** Grand mean training benefits for the three syllable sets and phoneme identification at the onset and coda. (**a**): pre–post-training changes in d-prime; (**b**): SNR, (**c**): normalized gain obtained from combining the training-related changes in d-prime and SNR into a single principal component. Error bars indicate the 95% confidence intervals.

Consonants in the onset location were identified with higher accuracy than the coda (t(51) = 7.0, $p < 0.0001$) and required a lower SNR (t(51) = 7.90, $p < 0.0001$). However, 'consonant set' × 'syllable type' interactions were found for d-prime (F(2,100) = 103, $p < 0.0001$) and SNR (F(2,100) = 79.2, $p < 0.0001$), which were caused by a stronger contrast between the onset and coda consonants in the C3 set than in the C1 or C2 (Figure 5c). Thus, better identification of the onset consonants was predominantly driven by the syllables in the voiced C3 set.

The ANOVA for d-prime revealed a 'session' × 'consonant set' interaction (F(2,100) = 18.8, $p < 0.0001$, caused by a larger pre–post increase in accuracy for C1 than for C3 (t(102) = 4.92, $p < 0.0001$), as well for C2 compared to C3 (t(102) = 3.27, $p = 0.0001$), but no difference between C1 and C2. An absent 'session' × 'syllable type' interaction suggested similar d-prime increases under the onset and the coda conditions. Additionally, the ANOVA for SNR revealed a 'session' × 'consonant set' interaction (F(2,100) = 8.09, $p = 0.0006$). A training-related SNR improvement of 2.2 dB for C2 was greater than the 1.0 dB change for C3 (t(102) = 3.06, $p = 0.003$). For SNR, a 'session' × 'syllable type' interaction (F(1,50) = 8.10, $p = 0.006$) was caused by a steeper SNR decrease for the onset than for the coda consonants (t(102) = 2.12, $p = 0.037$).

Despite the pronounced differences in d-prime and SNR levels for the different stimuli, the gain measure, combining SNR decrement and d-prime increment with similar weights, was not different between the C1 and C2 sets for onset and coda consonants (Figure 5). Only for C3 was the training-related gain was smaller. Quantitative analysis with an ANOVA for the gain revealed the effect of a 'consonant set' (F(2,100) = 19.1, $p < 0.0001$), however, no effect of 'syllable type' and no 'syllable type' × 'consonant set' interaction.

### 3.5. Effects of Consonants within a Set

Although the consonants within a set were phonetically similar, the participants' identification performance was noticeably variable between the consonants. Bar graphs in Figure 6 visualized the grand mean d-prime values for the consonants within the three stimulus sets, onset and coda location, and changes between the pre- and post-training tests.

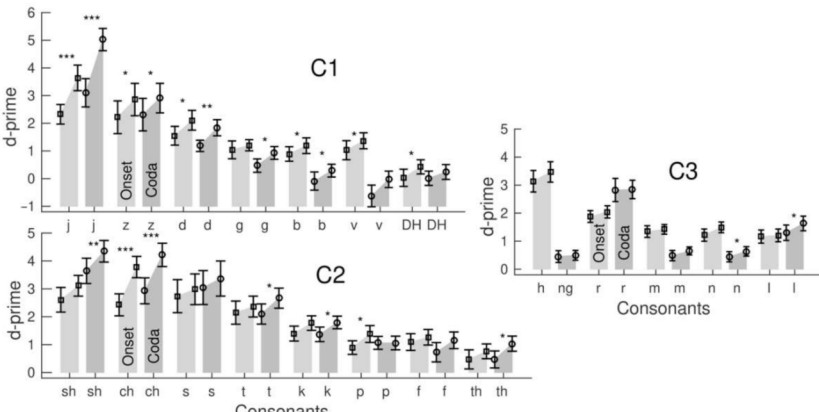

**Figure 6.** Group-mean d-prime for the individual consonants of the C1, C2, and C3 sets for the syllable test before and after training. The asterisks denote significant pre–post-training differences *: $p < 0.05$, **: $p < 0.01$, ***: $p < 0.001$ (paired *t*-test, FDR corrected); error bars indicate the 95% confidence intervals.

Separately for the stimulus sets and onset or coda location, repeated-measure ANOVAs for d-prime were performed with the factors 'session' and 'consonants' (Table 2). The ANOVAs revealed the effects of 'consonants' for all six conditions ($p < 0.0001$), indicating a different accuracy for the consonants within each set. For example, in the C1 coda condition, a low d-prime of 0.11 suggests that the stop consonant /b/ was challenging to identify. In contrast, a high d-prime of 4.1 reflected the easier identification of the affricative /j/ in the same set. Interactions between the 'session' and 'consonants' occurred for the onset and coda conditions in the C1 and C2 sets but not in the C3. For example, the d-prime increase of 0.22 was small for /DH/ in the C1 coda condition but as large as 1.9 for /j/ in the same set. The d-prime increases within the C3 set were small, with the most significant value of 0.32 for /l/ in the coda condition.

Easy-to-identify consonants with high initial accuracy showed a trend for greater d-prime gains. However, the correlation between the d-prime gain and pre-training d-prime varied across consonant sets and was strongest in the voiceless C2 set for the onset ($R^2 = 0.52$, F(1,51) = 54.0, $p < 0.0001$) and coda ($R^2 = 0.43$, (F(1,51) = 38.2, $p < 0.0001$). Pre-training accuracy was high for the affricatives /ch/ or /sh/, and the d-prime increase for these consonants was large, whereas the d-prime increase was small for more difficult-to-identify consonants. The same correlation for voiced consonants was weaker because of a noticeable d-prime increase for some more difficult-to-identify consonants. This was especially the case for the C1 coda ($R^2 = 0.33$, F(1,51) = 25.4, $p < 0.0001$), where consonants with a lower d-prime such as /b/ or /v/ significantly increased. However, the d-prime increase within C1 was most prominent for the affricative /j/. The correlation for the C3 coda was the weakest ($R^2 = 0.30$, F(1,51) = 21.0, $p < 0.0001$) because of no d-prime increase for the easiest identifiable consonant /r/.

**Table 2.** Repeated-measure ANOVAs for d-prime for C1, C2, and C3 consonant sets for onset and coda condition with the within-subject factors 'session' (pre- and post-training test) and 'consonants' (C1: n = 7, C2: n = 8, C3: n = 5). Significant *p*-values ($p < 0.05$) are in bold.

| Effect | d-Prime | | | | | |
|---|---|---|---|---|---|---|
| | **C1 Onset** | | **C2 Onset** | | **C3 Onset** | |
| Session | $F(1,51) = 31.3$ | **$p < 0.0001$** | $F(1,51) = 18.1$ | **$p = 0.0001$** | $F(1,51) = 7.26$ | **$p = 0.0095$** |
| Consonants | $F(6,306) = 58.0$ | **$p < 0.0001$** | $F(7,357) = 57.0$ | **$p < 0.0001$** | $F(4,204) = 101$ | **$p < 0.0001$** |
| Session × Consonants | $F(6,306) = 4.86$ | **$p = 0.0001$** | $F(7,357) = 3.71$ | **$p = 0.0007$** | $F(4,204) = 1.25$ | $p = 0.293$ |

**Table 2.** *Cont.*

| Effect | d-Prime | | | | | |
|---|---|---|---|---|---|---|
| | C1 Coda | | C2 Coda | | C3 Coda | |
| Session | $F(1,51) = 59.1$ | $p < 0.0001$ | $F(1,51) = 32.2$ | $p < 0.0001$ | $F(1,51) = 6.77$ | $p = 0.012$ |
| Consonants | $F(6,306) = 171$ | $p < 0.0001$ | $F(7,357) = 85.7$ | $p < 0.0001$ | $F(4,204) = 188$ | $p < 0.0001$ |
| Session $\times$ Consonants | $F(6,306) = 11.5$ | $p < 0.0001$ | $F(7,357) = 3.33$ | $p = 0.002$ | $F(4,204) = 1.22$ | $p = 0.30$ |

### 3.6. Training Benefit Versus Hearing Loss

The ANOVA with 'PTA' as a between-participant factor (Table 1) showed an effect of 'PTA' on d-prime ($F(1,50) = 48.3$, $p < 0.0001$) and SNR ($F(1,50) = 68.1$, $p < 0.0001$). These strong effects were expected because of the correlation between the SIN and hearing loss; participants with more severe hearing loss required a larger SNR. However, a 'PTA' $\times$ 'Session' interaction was only significant for d-prime ($F(1,50) = 9.41$, $p = 0.0035$) but not for SNR ($F(1,50) = 2.75$, $p = 0.10$). Moreover, there was no effect of 'PTA' on the normalized gain ($F(1,5) = 0.90$, $p = 0.35$). Thus, despite variable accuracy and SNR depending on the individual hearing loss, the overall gain measure indicated that participants within a wide range of hearing abilities could benefit from the training.

The ANOVA revealed a 'PTA' $\times$ 'Consonant Set' interaction for SNR and 'PTA' $\times$ 'Syllable Type' interactions for d-prime and SNR. How d-prime and SNR depended on PTA was studied by stratifying participants into three PTA groups.

The relationship between the accuracy level and hearing loss is illustrated for the C1 coda condition with a scatter plot of individual d-prime versus PTA measures and linear modelling in Figure 7a: d-prime was negatively correlated with PTA (pre-training: $R^2 = 0.45$, $F(1,50) = 41.9$, $p < 0.0001$, post-training $R^2 = 0.29$, $F(1,50) = 20.4$, $p < 0.0001$). Generally, the accuracy was lower with a larger hearing loss. The analysis of covariance showed that the slopes were only marginally different ($F(1,100) = 2.86$, $p = 0.094$); Figure 7a suggests that the d-prime increase was slightly larger in individuals with a higher rather than a lower PTA.

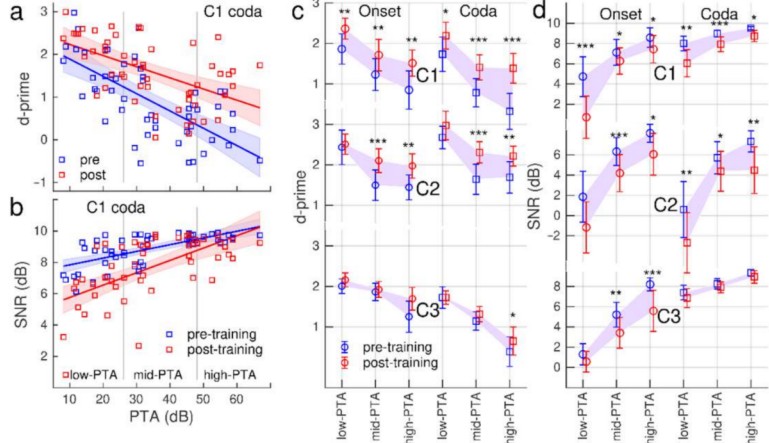

**Figure 7.** Relationship between syllable recognition and hearing loss. (**a**): Individual accuracy measured with d-prime pre- and post-training phonemes at the coda position of syllables from the C1 set. Blue and red lines indicate linear models for the relationship between d-prime and PTA. The shaded areas are the 95% confidence regions for the linear model. (**b**): Individual SNR values for pre–post-training tests and linear modelling with PTA. (**c**): Group mean d-prime measures for the six stimuli in three subgroups low-PTA (n = 13 lower quartile of PTA), mid-PTA (n = 26 centre quartiles of PTA), and high-PTA (n = 13, upper quartile of PTA). The shaded areas indicate the training-related increase in d-prime. Error bars indicate the 95% confidence intervals. (**d**): Group mean SNR measures for the six stimuli.

Linear modelling of the relationship between the mean SNR and hearing loss is illustrated in Figure 7b. Participants with a larger hearing loss required a larger SNR before the training ($R^2$ = 0.44, F(1,50) = 41.0, $p$ < 0.0001) and after the training ($R^2$ = 0.38, F(1,50) = 31.1, $p$ < 0.0001). However, the slopes were different (F(1,100) = 5.29, $p$ = 0.024); the SNR decrease was stronger with less severe hearing loss.

How the d-prime and SNR levels varied with hearing loss is summarized with group-mean values in the three PTA subgroups for all stimulus conditions in Figure 7c,d. Consistently across the stimuli, accuracy was lower with more severe hearing loss. However, d-prime increased between pre- and post-training tests. The d-prime increase was found to be more prominent in the high-PTA and mid-PTA groups compared to low-PTA (Figure 7c) consistently with the interactions between 'PTA' and 'stimulus set' and 'syllable type' found in the ANOVA (Table 1). The SNR requirement was generally higher with a more severe hearing loss (Figure 7d). Participants performed at lower SNR after the training and the SNR change was homogeneous across the hearing loss groups; the ANOVA did not show a 'PTA' × 'Session' interaction for SNR.

### 3.7. Training Transfer on Sentence Recognition in Noise

The grand mean training effect of 0.31 dB for the QuickSIN was insignificant (t(51 = 1.26, $p$ = 0.21). Similarly, no training effects were found for the BKB-SIN test.

### 3.8. Effects of MoCA Scores on Training Benefit

We investigated the effect of cognitive decline on training benefits by including the MoCA scores as a between-participant factor to the repeated-measure ANOVAs for d-prime and SNR. MoCA did not impact d-prime. However, for SNR, a 'MoCA' × 'session' interaction (F(1,50) = 6.41, $p$ = 0.015) was observed because the participants with higher MoCA scores benefitted most from the training as indicated by a larger decrease in SNR after the training ($R^2$ =0.13, F(1,51) = 7.31, $p$ =0.045). We excluded the participant with the outlier MoCA score of 18 from this analysis.

## 4. Discussion

We investigated how older adults with a wide range of hearing abilities could benefit from adaptive listening training that resembled realistic everyday listening situations. Generally, all participants could perform the training and improve their competence levels. For participants with relatively good performance, the level of difficulty was adapted through the staircase procedure, and these participants improved their listening at a lower SNR. For participants with poorer performance, the adaptive procedure adjusted the SNR to a higher level, still relevant for a realistic listening situation. These participants improved in accuracy but they required a high SNR.

Training benefits were more prominent for the voiced and voiceless stop and (af)fricative consonants. Although identifying more difficult consonants improved after the training, the most significant gain emerged for the easier identifiable consonants. The individual benefit differed depending on peripheral hearing loss and SIN deficits. Participants with a hearing loss improved accuracy but still required a high SNR. In contrast, people with better hearing generally had a better SIN understanding and performed with high accuracy at lower SNR. Noticeably, the participants with severe SIN deficits improved syllable identification in noise. However, an overall transfer effect on sentence recognition was not detectable for the short-term training.

We examined whether training with highly confusable consonants in a closed-set, forced-choice paradigm was beneficial to improving syllable identification in noise. The literature informed the new training paradigm showing training benefits for demanding tasks [52] using similar material for training and testing [52,79]. Thus, we related the results of the syllable test to perceptual learning. One caveat is that the familiarization effects, which could occur between repeated tests, may have exaggerated the training outcome.

However, the stimuli were randomly selected from a large set of 2400 syllables. Thus, it is unlikely that participants became familiar with specific syllables.

Our findings were consistent with a study by Woods et al. [41] which was, to our knowledge, the only training study that used the same CVC syllables. The authors adjusted the SNR for each consonant to maintain accuracy at a constant level in an open-set paradigm. Moreover, the listeners typed their responses. Despite these differences, participants in our study gradually decreased the SNR and increased accuracy for most syllable sets.

The pre–post-training syllable tests revealed performance changes comparable to those throughout the training. However, the training benefits varied for different consonant sets. The gains for the syllable sets with voiced and voiceless stop and (af)fricative consonants were significant, while the improvements for nasal and glide consonants were minor. One reason could be that presenting the syllables in smaller constant sets impacts the adaptability or transfer of certain distinctive information on other speech cues. For example, the long-lasting high-frequency noise in /ch/ or /j/ [70] provides sufficient information to easily distinguish those speech cues from others. However, other acoustic-phonetic features are less concise. For example, Woods et al. [41] found that confusion between consonants such as /n/-/m/-/ng/ persisted even after training. These findings might also explain the identification differences between the individual consonants within a set. In most training studies, participants improved for consonants identified before the training as more difficult-to-identify and showed better correct identification [38] or an elevated consonant threshold [41]. Noticeable improvements for those consonants with initially low accuracy in our study confirmed these findings. However, we observed the most significant gains for the initially easiest-to-identify consonants, the (af)fricatives /j/, /ch/, and /sh/; consonants also shown in other studies as easily identifiable [27,38,41]. Thus, the adaptive training of stimuli sets with phonetic similarities seemingly improves phoneme identification for stimuli that are difficult and easy to identify. The task difficulty was balanced between easy and challenging to identify stimuli but remained achievable for the trainee, which is crucial for successful learning [80].

Factors other than the acoustic properties of the stimuli could have impacted the reported findings, like the frequency of stimulus occurrence. For example, in a training study with older adults [81], the training benefits were most prominent for those participants who were trained with words frequently used in the American English. Participants in our study gained most significantly from consonants such as /j/, /s/, or /t/, which are more frequently occur in the English language [82,83] compared to less-frequent consonants such as /ng/. Occasionally, participants had difficulties associating a sound to the orthographically presented syllable on the screen, especially for uncommon symbols such as 'DH' for /ð/. Some participants reported that they sometimes forgot how 'DH' should 'sound'. Therefore, challenging sound-orthography congruence for some syllables might have been another factor causing the difference between the consonant sets. Several amendments to the training paradigm could facilitate or enhance the perception of specific speech cues. For example, target sound presentation in the quiet, visualization of lip-movement during articulation (e.g., as for /m/ versus /l/), or the acquisition of sensorimotor skills, e.g., inserting a 'reading-aloud' task.

Contrary to our prediction, training benefits were more pronounced for consonants in the onset than coda syllable position, expressed in a steeper SNR decrease. Focusing on the beginning or end of a nonsense syllable might favour easier onset identification. A previous study using an open-set paradigm [41] did not find training differences in the syllable position. Thus, further research is necessary. Unfortunately, previous studies rarely emphasized training differences in different syllable positions in older adults with age-related hearing loss.

Our adaptive training paradigm was aimed at including people with a wide range of capabilities. The SNR was adapted to individual hearing abilities using a modified staircase procedure. Some observed performance differences may be specific to procedural details. For example, we limited the SNR range to resemble a typical real-life listening

situation. However, people with more severe hearing loss required larger SNRs than their normal-hearing peers and performed at ceiling SNR. Older adults with a hearing loss need a 30 dB larger SNR range for consonant identification than adults with better hearing [35]. For example, the SNR required to identify onset consonants ranged from 6 dB for /d/ up to 38 dB for /DH/ for listeners with hearing loss [49]. The set C1 in our study contained these two consonants, leading to performance variability. When presenting the stimuli in random order, the trial-by-trial changes in SNR step size were not always sufficient to adapt to SNR requirements. Consequently, specific consonants were more affected by the noise, impacting differently on participants' response selections. Participants reported that they found the overall training challenging but very motivating. A high motivational level seems beneficial for successful training [75,76].

Although the SNR varied across stimuli, training benefits could be observed in participants accepting a higher noise level (i.e., lower SNR) and improving their accuracy in the mean across an experimental block. Numerically, the training-related changes in SNR and accuracy are specific to the procedure and may differ from outcome measures reported in other studies. However, the most important advantage of adaptive training was accommodating a wide range of hearing abilities. The paradigm allowed each listener to train and improve at their individual 'edge of competence'.

The training benefits depended on strongly correlated hearing loss and SIN loss. About two-thirds of the listeners with less hearing loss also had slight SIN loss and vice versa. However, some individuals differed in performance and training-related improvement from their peer groups. For example, one participant had more extensive hearing loss, but good SIN recognition and the training gains were comparable to participants with a lower PTA. Another individual with low PTA showed high SIN deficits, and training improvement was similar to the high-PTA group. Conclusively, the paradigm sensitively incorporates the requirement according to the large variability of SIN performances within the aging population.

Generally, people with higher MoCA scores benefited more from the training. However, even participants with mild cognitive impairments could perform the training and improve their abilities. For example, the participant with the lowest MoCA score of 18 increased accuracies comparably to participants with higher scores. A more extensive cognitive test battery is required besides the single MoCA score to better disentangle cognitive factors' effects.

Current evidence for a generalization of SIN training is weak [3,84]. Additionally, we did not find an overall improvement in sentence recognition in noise. However, the training was particularly beneficial for those participants with the initially most unsatisfactory performances in SIN recognition, which is comparable to previous findings [56]. Moreover, the gains were observed for QuickSIN, where the listener's performance relies more on high-frequency speech cues than contextual information provided in BKB-SIN [63]. Thus, training high-frequency speech cues in the form of complex syllables within background noise might enhance syllable processing performance for some individuals, even when presented in higher-level speech recognition contexts such as a sentence.

In conclusion, the adaptive procedure of our paradigm captured individual differences and adjusted the level of difficulty. It allowed each listener to train at their level of competency and balanced differences in stimuli difficulty. Therefore, the training paradigm accommodated the heterogeneous aging population by accounting for the large variability of hearing loss, SIN deficits, and cognitive capabilities in older adults. Moreover, with the increasing focus on patient-centered and self-management approaches in adult auditory rehabilitation, individualized training approaches provide an excellent opportunity to support new technologies such as e-health in clinical practice.

**Author Contributions:** Conceptualization, A.S.; methodology, software, and formal analysis, A.S. and B.R.; investigation, A.S.; writing—original draft preparation and editing, A.S. and B.R.; visualization, A.S. and B.R.; supervision, B.R.; project administration, A.S.; funding acquisition, A.S. and B.R. All authors have read and agreed to the published version of the manuscript.

**Funding:** This research was funded by the Canadian Institutes of Health Research (CIHR), grant number PJT162215 to B.R., and a post-doctoral fellowship to A.S. from the German Research Foundation.

**Institutional Review Board Statement:** The study protocols conformed to the World Medical Association Declaration of Helsinki (World Medical Association, 2013) and were approved by the Research Ethics Board at Baycrest Centre for Geriatric Care (REB 18-35).

**Informed Consent Statement:** Informed consent was obtained from all participants involved in the study.

**Data Availability Statement:** The data presented in this study are available upon request from the corresponding author. The institutional policy is that a formal data-sharing agreement is required to share the data with outside researchers.

**Conflicts of Interest:** The authors declare no conflict of interest.

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
