# Peer review of "Adaptive Syllable Training Improves Phoneme Identification in Older Listeners with and without Hearing Loss"

_audiolres, doi:10.3390/audiolres12060063_

Round 1
Reviewer 1 Report
Adaptive syllable training improves phoneme identification in older listeners with and without hearing loss
Major comments:
The manuscript presents a training study where a decent sample (N=52) of older listeners (60 to 90 years of age) with varying degrees of hearing loss were trained to identify syllables. There were 5 training sessions over the course of 2 weeks, with a pre- and a post-training test to evaluate the efficiency of the paradigm. The authors considered several aspects of the training protocol, including maximizing the discriminative power between phonetically similar syllables (in different sets) to sharpen the identification process, and to push participants toward difficulty levels where they can remain engaged and face challenges while not being discouraged. And generally speaking, the more diverse the population, the more difficult this exercise becomes. So, with an ageing population having varying ranges of PTA, there’s a lot of heterogeneity to cover. Thus, I wish to congratulate the authors on their effort. However, the take-home message could be made clearer or stronger: currently, it stands as “training paradigms are available to try and help older listeners with/without HL in improving their skills for speech recognition in noise”. And I think the paper could take on a more ecologically relevant perspective to better explain why this is needed in our society. The first 5 lines of the intro are summarizing the rationale (lines 33-37) and this is what could be considerably expanded and later discussed.
Apart from that, I don’t see any major flaw in the design or analysis of the data, with the exception of one famous issue that concerns all training studies, namely the Regression toward the mean. So, let me briefly explain the problem in a nutshell. A person doing very badly on a first test is likely to remain poor on a second test, because there are good reasons (replicable reasons) why they did poorly, for example the degree of their HL. But critically, they might not perform as bad on re-test as on the first time, because there was an element of randomness, i.e. they were especially on a bad day the first time. Similarly, a person doing really well on the first test is likely to keep performing fine but presumably not as good as the first time. In other words, the extremes regress back to the middle (and the new extremes are more likely to have come from the previous middle). This is why when one compares the magnitude of a gain between pre- and post-training, it is often found that the people who gain the most are the poorest performers, while the best performers gain the least. And essentially, that does not say anything about training; it is purely a statistical feature.
Now, as far as I can tell, few aspects of the current analysis are invalid because of this issue. Some findings do fall in this pattern (e.g. line 736-738) while other trends run in contradiction to it. But for a study on training, I think it would be simply important to remind the reader of this notorious problem, and emphasize parts where this issue could have come into play but for some reason did not materialize itself fully. As long as the authors acknowledge this potential concern and discuss its applicability, I have only minor comments throughout the text (below).
Comments throughout the manuscript:
Line 55-57: this literature in general, and here the rationale underlying articulation indices, could be better explained.
Line 59: it’s odd to state hearing loss as a cause; SIN deficits are just one facet of hearing loss in general. They are the consequence of a number of peripheral degradations, in addition to central factors.
Line 73-74: confusions should always be easier for stimuli matched in phonetic similarity, so the term “even” is confusing here. It seems to create an opposition where there’s none.
Line 84-85: again, the causality is odd here. Behavior is necessarily the output, so neural reorganization is the driver of the performance changes, not the other way around.
Line 92-93: Are you saying that it’s essential to have an active task for the training benefits to arise quickly? Maybe clarify the timeline difference expected between active and passive tasks.
Line 104-108: both of these studies are critical to the rationale for the study. I recommend spending a bit more time on them. In [35] it might be simpler to state the increase in performance or the lowering of SNR, but not both simultaneously. Also, clarify the type of noise and the characteristics of the sample (age range, duration of HA use etc..). In [38], the generalization to untrained voices is very important; clarify how many voices were in the training set, and how many in the test set.
Line 109-110: Be careful with this statement, it sounds a bit like “the difference post – pre training was larger for individuals with the poorest pre-training performance” This is generally considered a fallacy, referred to regression toward the mean.
Line 115-118: To some degree, this idea runs in contradiction with the notion that training is optimal when pushing participants to their limit (e.g. line 43-44). Can you expand on why the generalization would occur more for easy tasks then?
Line 118-120: perhaps better placed after line 115, but overall I suggest the authors to expand on the role of stimulus variability which is essential for generalization but might lead to weaker benefits.
Line 128: given [41] and [56], why not expecting the same as the literature?
Line 137-139: conditional tense instead?
Line 150-151: sure, but that shouldn’t be a reason not to do it. Variability is essential, even though it poses some practical constraints.
Line 159-160: You lost me here. Are you using a staircase procedure or not?
Line 161: No group have been defined yet. Are you speaking about within-subject differences?
Line 176: This is a pity. Comparison between syllable and sentence recognition in noise would have been very interesting, especially in the light of generalization effects. But line 178 seems in contradiction. Clarify.
Line 229: The link is broken or outdated. Clarify whether these materials are publicly available.
Line 245: “more often misidentified than others despite their similarity” Do you mean “because of their similarity”?
Line 271: Clarify what’s a block: one of the 6 consonant sets?
Line 276: So, these tests were at least separated by a few hours from the first and fifth training session?
Line 281: you did conduct a staircase on SNR then, OK. Not sure why there was a contradiction earlier.
Line 285: the same syllable was presented again in case of a mistake; was it also presented with the same set of 3 orthographically presented alternatives?
Line 289-300: Figure 1d is very useful. Thanks.
Line 309: “8 reversals or a maximum of 10 minutes” It’s strange to use a timer for completion; I don’t see why this would ensure at least 80 trials.
Line 330/342: Taking the mean SNR across all trials will not lead to the expected 79.4% for the 3AFC, this is only true when extracting SNR at the reversal points. So, this is a bit unorthodox, pooling all responses together to extract Hits and FAs while they correspond to an ensemble of different SNRs. It’s not wrong and I can see why this approach is relevant in this context, but given hysteresis effects, Hits might be calculated from a slightly lower SNR than FAs. So, the d’ values are given at an approximate SNR. It’s fine but perhaps explain why such an approach is necessary, as opposed to standard staircases. In fact, I would have thought a Bayesian approach would be most useful.
Line 348-349: Replace “hearing loss” by “peripheral loss”. But it was mentioned earlier that PTA and MoCA were used as covariates, not between-sub factors. So, was it a mixed ANOVA or an ANCOVA?
Line 351-356: You lost me here. Clarify how this differs from the previous paragraph. Why are there now 6 rm-ANOVAs?
Line 363: it’s not clear where to see the 7.5 dB value in Fig2a. Also, please bring shaded areas to show +/-1 sd or 1 se around these curves.
Line 372: after “(Figure2)” shorten the wording “and this relationship was steeper for QuickSIN than BKBSIN”.
Line 377: I was searching for the same correlations with BKBSIN but I guess they are not plotted since they were not as clear. Is that right? Clarify why you chose one, or alternatively state the correlation between QuickSIN and BKB-SIN (which I presume exists), take an average between the 2, and base these correlation attempts to this average SIN score. This will also limit your Bonferroni adjustment (which I don’t believe was mentioned).
Line 379: that’s right, this is a problem of heteroscedasticity which is generally not trivial to address.
Line 392: Here, explain one more time why this volatility occurred.
Line 426-429: This is excellent. I wish this had been made clearer in the methods. Figures 3d and 3e are very useful.
Line 470: clarify Figure 5, bottom panel on the right
Line 472-481: refer to the appropriate figure/panels
Line 494-504: I now see what you stated earlier in the methods. But to some degree, this dilutes the main point. Delving into each consonant could be placed in an appendix to keep the main result simple and concise, with a brief mention that certain consonants reflected impressive gains (see Appendix xx for more details).
Line 505-517: These results are informative, but keep “regression toward the mean” in mind. As far as I can see, most of these trends run in contradiction to it, so it is quite intriguing. There’s something genuine here, but it would be useful to alert the reader that such analyses have a slippery slope. (See major comment).
Line 529: The fact that training benefits are independent of PTA is also puzzling, given “regression toward the mean”.
Line 543-544: same idea, if there’s an element of randomness, it is actually underestimating the differences observed here.
Line 556-560: This finding, however, seems to follow the expected pattern of larger gains in SIN for the people with the largest SIN loss to begin with.
Line 632: perhaps clarify why both effects are observed because of the nature / rules of the training
Line 639: This is the part I find disappointing, and it would be greatly consolidated by showing the sentence-in-noise data, not just the syllable-in-noise data. Is it really too big a dataset to incorporate in the same paper?
Line 647-648: I agree. But I find the emphasis on training / testing similarity more problematic (line 644). It reads as though the demonstration of the training efficiency is the crux of the exercise, whereas it should be about the ecological relevance of this training. From the perspective of the patients (and clinicians) in auditory/speech rehabilitation, you want to know whether a training confers benefits beyond the trained materials. Who cares if a test can be manufactured in a way that demonstrates benefits because it is close to the trained materials?
Line 668: typically a pattern expected from regression toward the mean.
Line 692: but the literature cited in the introduction, did it not go against this hypothesis, in line with the data. Was your prediction that benefits would be similar for onset and coda positions or that they would be more pronounced in the latter?
Line 698-712: Yes, this is a difficult paradigm to get right. It has to push participants to their limit and yet be sufficiently versatile to accommodate different consonants whose psychometric functions lie in very different ranges of SNR. I can only think of smaller subsets but the smaller the subset, the less generalizable it becomes and so, it’s a catch-22. I appreciate the authors’ effort and design, despite these limitations.
Line 720-728: I find it strange that the MoCA was not able to help understand these counter-examples (at least for d-prime).
Line 731-732: but if this individual was comparable, why was he excluded from the analysis (line 567)?
Line 736-738: typically a pattern expected from regression toward the mean.
Line 744-751: at last, some words to back up the ecological relevance of the study. I think this should have been acknowledged earlier, in the intro but perhaps also in the abstract (instead of focusing on the intricacies of perceptual learning).
Reviewer 2 Report
Overall, the studies were well done and enough details were presented for each experiment. However, the manuscript is quite lengthy and I even lost interest to read the entire paper at some point. I would recommend the authors should try to present the most valuable information in future revisions.
The introduction should be shortened and it should focus on reviewing the most relevant studies that have been performed.
The results section: please consider moving some figures and results to an appendix.
References: please consider reducing the numbers of paper cited to less than 40 if possible.
Fig 1. what is inside the empty boxes in the flow chart?
Fig 3: why the sum of a few rows in the confusion matrix is over 1.0?
Reviewer 3 Report
The manuscript describes a study asking whether training can improve speech recognition in older listeners. The subject population is well known to have difficulty understanding speech, especially in unfavorable listening conditions. The study addressed an important issue, and that is its strength. The value and effectiveness of training has been studied by others, of course; the approach here might best be described as an attempt to refine a particular training strategy. Given that limited goal, the experimental design, the methods of analysis, and the presentation of results all seem to me to be more complicated than would have been necessary. Similarly, the introduction seems too long, because it reviews facts that are already familiar to the specialists who seem most likely to read a paper on this specific topic. Figure 1 presents details - waveforms, spectrograms, the spectrum of speech-shaped noise - that aren't necessary or useful for a report about the effectiveness of training. Overall, I think the authors' points could be made more effectively with a shorter, more focused presentation.
Round 2
Reviewer 2 Report
accept as it is.